# Stark Effect for Donors in Rolled-Up Quantum Well

**DOI:** 10.3390/mi14071290

**Published:** 2023-06-24

**Authors:** Luis Francisco Garcia Russi, Ilia D. Mikhailov, Ruthber Antonio Escorcia Caballero, Jose Sierra Ortega, Gene Elizabeth Escorcia Salas

**Affiliations:** 1Escuela de Física, Facultad de Ciencias, Universidad Industrial de Santander, A. A. 678, Bucaramanga 680002, Colombia; lfragar@gmail.com (L.F.G.R.); mikhail2811@gmail.com (I.D.M.); 2Grupo de Investigación en Teoría de la Materia Condensada, Universidad del Magdalena, Santa Marta 470001, Colombia; ruthberescorcia@unimagdalena.edu.co (R.A.E.C.); geneescorcias@unicesar.edu.co (G.E.E.S.); 3Grupo de Óptica e Informática, Departamento de Física, Universidad Popular del Cesar, Sede Hurtado, Valledupar 200001, Colombia

**Keywords:** rolled-up quantum well, donor impurity, binding energy, external electric field, Stark effect

## Abstract

We calculate energies of shallow donors confined in a rolled-up quantum well in the presence of the electric field by solving numerically the Schrödinger equation in natural curvilinear coordinates. It is found that the curves of density of states (DOSs) are very sensitive to the variation of the donor position, the geometry of the spiral and the applied electric field value. Novel results for dependencies of donor’s dipole moment and its polarizability on the electric field strength and its orientation, for different donor positions are presented. Additionally, we found that the anisotropic Stark effect of the first order provides in this structure a dependency of the polarizability on the external electric field in a spike-like shape, giving rise to a sharp variation of the dipole moment.

## 1. Introduction

Electronic properties of a 2D semiconductor and a semiconductor bulk differ substantially. As the semiconductor film’s thickness reduces to a nanometric range, the ratio between the number of surface and bulk atoms is increased substantially, providing a drastic variation of its electronic properties. The semiconductor quantum well (QW), a layer of nanometric thickness, sandwiched between two layers of a different semiconductor material, today enjoy widespread use in different optoelectronic applications. Progress achieved in recent years in the manufacturing of nanostructures has allowed for fabricating layers with topologically nontrivial surfaces, such as a twisted single crystalline NbSe3 ribbon in the form of a one-sided Mobius strip [1] or rolled-up quantum wells (RUQWs), in the form of spirals [2,3,4,5,6,7,8].

The RUQW is a structure consisting of a semiconductor layer of nanometer thickness that curls up after being released from the substrate as a result of the relaxation of elastic stresses. Both sides of the layer are cylindrical surfaces, whose cross sections have the form of spirals. The curvature of this structure and its optoelectronic properties depend strongly on the thickness of the layer and the lattice mismatch of the substrate and the layer. Due to the existence of a variable curvature along the spiral cross section, electronic properties of RUQWs have characteristic features that distinguish them both from planar quantum wells and nanotubes. Rolling up a thin semiconductor film into a rolled nanotube is especially interesting because it has a unique pattern similar to that of a cylindrically symmetrical crystal. In the last two decades, these rolled-up semiconductors have sparked great attention and been widely used as optical resonators [9,10,11,12,13,14,15,16] to realize laser structures [17,18,19], and the alteration of an energy band in self-rolled nanomembranes containing $GaAs/Al_{0.26}Ga_{0.74}As$ quantum wells (QWs) has been investigated [20]. Spectral analyses reveal that for small strain change in structures with different curvatures, the optical transitions and the light emission intensity of the GaAs QW layer are essentially influenced by the strain evolution. In addition, the first-order Stark effect revealed in these structures opens an opportunity for the fabrication of high-performance rolled-QW infrared photo-detectors.

Wide scopes for applications of RUQWs heterostructures as optical materials for quantum information technology are discussed in ref. [21], where the authors found an essential modification of the band structure of InGaAs semiconductor quantum wells by changing their geometrical configuration. Particularly, the photoluminescence demonstrates a strong energy shift of the valence-band states in comparison to flat structures due to an inversion of the heavy-hole and light-hole states in a rolled-up InGaAs quantum well, which, in addition, is conducive to a strong change in the optical selection rules for the rolled-up quantum wells. Previously, the relation between the photoluminescence characteristics and curvature for rolled-up GaAs quantum wells was studied in ref. [22]. It was revealed that the room-temperature photoluminescence (PL) characteristics of RUQWs show dramatic PL intensity enhancement compared to their planar counterparts. A systematic shift of the PL peak position as a function of the tube curvature, attributed to the strain, which induces band structure change, was established. These results reveal that 3D flexible nanomembranes can have unique properties required for flexible electronics, photovoltaics, and photonics. The RUQW 3D self-assembly offers a new path to high light-to-electricity conversion efficiency in photodetectors, solar cells and light-emitting diodes [22]. Tuning the photoluminescence characteristics with curvature for rolled-up GaAs quantum well microtubes was established later [23]. Effects of voltage and temperature on photoelectric properties of rolled-up quantum well nanomembranes were analyzed in Ref. [24].

Experimental progress in the fabrication of RUQWS generated interest in a comprehensive theoretical understanding of the dynamics of current carriers in rolled-up nanofilms. As it was shown in Refs. [22,25], a formal description of the dynamic of carriers confined inside quasi-two-dimensional film becomes well defined only when their confinement inside a curved 2D manifold is treated in the framework of the adiabatic approximation (AA) applied to the limiting case of a particle in a strongly anisotropic 3D manifold. In this approximation, a shallow donor placed inside a RUQW has a spectrum similar to a hydrogen-like atom in the curve 2D space, with the excessive electron in the ground-state configuration located mainly at the same wing of the spiral. Such a configuration can be unstable with respect to a possible jump of the electron toward one of the adjacent wings under an external electric field applied perpendicularly to the spiral’s wings. Therefore, one can expect unusual spectral features of RUQWs, related to a nonlinear and anisotropic Stark effect.

In this work, we analyze theoretically the effect of a possible increase in the probability of the carriers tunneling between adjacent spiral wings in the presence of the external electric field, enabled by the special topology of the RUQW, which could be a possible source of other novel useful properties. In order to analyze this effect, one should solve a boundary value problem for the Schrodinger equation inside of a domain, which represents a waveguide formed by a semiconductor tape rolled up around the *z*-axis. In this case, when the tape thickness is essentially smaller in comparison to the separations between wings, the motion across the waveguide is considerably faster than along it, which opens a chance for the application of the well-known adiabatic approximation (AA) to separate the corresponding coordinates of the electron’s motions along and across the spiral waveguide. Such an approach was used earlier in references [25,26] for analyzing the energy spectrum of the free electron confined in infinite-barrier RUQW in the form of the Archimedean spiral. It is found an effective one-dimensional Schrödinger equation that describes in curvilinear coordinates the slow motion of the electron inside the waveguide. In this paper, we use a similar formalism to derive an effective wave equation in curvilinear coordinates corresponding to the low-lying energy levels of the electron bound to the shallow donor. Unlike a similar problem for a free electron, the movements of the electron bound to the donor at Z-direction and along the spiral no longer can be separated, and the corresponding effective wave equation is two-dimensional. In our numerical work, to solve it, we use the double Fourier series expansion method.

Our calculations for donors with different locations show that energies shifting in the presence of the external electric field depend heavily on the geometry of the spiral, the strength, and the orientation of the applied field, inducing in some particular cases permanent dipole moments. We believe that our numerical procedure gives an efficient tool for the study of impurity-related phenomena in RUQWs. The article is organized as follows. In Section 2, we describe the formalism, in which we solve natural coordinates for electron and donor wave equations in RUQW and determine low-lying energies as functions of the external electric field. In Section 3, we present the results of calculations and discussion of the Stark effect for the shallow donor confined in RUQW. Finally, the conclusion is drawn in Section 4.

## 2. Theoretical Model

To analyze the features of the RUQW electronic properties related to their curvature, let us consider a model of the homogeneous rolled-up QW represented schematically in Figure 1, flat in the z-direction and rolled-up in the x-y plane. We assume that the height of the QW barrier is high enough so that the probability of jumps to the neighboring RUQW wings would be essentially lower than displacement along the spiral. It allows us to reduce the analysis to the movement of the leaving donor electron to study its displacements only along a helical wave guide. In addition, if the QW thickness is very small, for the electron movement, transversely, the waveguide is essentially faster than along it, and one can use an advantage of the well-known adiabatic approximation [25] to separate coordinates corresponding to these two degrees of freedom. For this purpose, it is appropriate to describe the RUQW geometry in natural coordinates given by unit vectors (${\vec{e}}_{s},{\vec{e}}_{h},{\vec{e}}_{z}$) shown in Figure 1, which change from point to point in the space. The vector ${\vec{e}}_{s}$ describes the tangential direction for the curvilinear coordinate *s*, which measures the arc length from the beginning point of the spiral with s=0 up to the final point, where $s=s_{F}$ ($0<s<s_{F}$). The vector ${\vec{e}}_{h}$ describes the normal direction, and the corresponding coordinate *h* measures the distance from the exterior border of the QW along the normal. Lastly, the vector ${\vec{e}}_{z}$ describes the vertical direction with corresponding to the coordinate *z* that gives the distance from the bottom of the structure (0<z<L). Below, we describe the dimension of the RUQW structure by means of three parameters: the spiral length $s_{F}$, the thickness *d* and the height *L* of the QW. In addition, we assume that in the adiabatic limit (d→0), the normal coordinates both for the donor and the electron can be accepted as equal to h=0, and their corresponding position vectors can be defined as follows ${\vec{r}}_{D}=\left(S_{D},0,L/2\right)$ and ${\vec{r}}_{e}=\left(S,0,z\right)$. In our model shown in Figure 1b, an external electric field is applied in the XOY plane along a line forming an angle θ with the *X* axis.

Here, following refs. [25,26], we concentrate on particular rolled-up nanostructures, assuming that cross sections of the RUQWs along the XOY planes are spirals given in polar coordinates by the Archimedean equation r(φ)=a+bφ/2π; with $0<\varphi <\varphi_{F}$. The arc length *s* and the curvature radius $R_{s}$ of this curve depend on the polar angle as follows:(1)$$s\left(\varphi \right)=\int_{0}^{\varphi }R_{s}\left(\varphi \right)d\varphi ;\;R_{s}\left(\varphi \right)=\sqrt{{\left(a+b\varphi /2\pi \right)}^{2}+{\left(b/2\pi \right)}^{2}}$$

In our numerical work, we first found the dependence of the polar angle on the arc length $\varphi =\tilde{\varphi }\left(s\right)$ by using the inverse cubic spline-interpolation procedure. Once such dependence is found, then one can define similar dependencies for the Cartesian and polar coordinates of the electron and donor positions:(2)$$\tilde{r}\left(s\right)=a+b\tilde{(}\varphi \left(s\right))/2\pi ;\;\tilde{x}\left(s\right)=\tilde{r}\left(s\right)\cos\varphi \left(s\right);\;\tilde{y}\left(s\right)=\tilde{r}\left(s\right)\sin\varphi \left(s\right)$$

Below, we use dimensionless units, the effective Bohr radius $a_{0}^{*}$ for distance, and the effective Rydberg, $R_{y}^{*}$, for energies. In these units, the effective 2D potential energy of the donor confined in a RUQW and the electron–donor separation are given by the following expression [25,26,27]:(3)$$\begin{array}{c} V\left(s,z\right)=\frac{\pi^{2}}{d^{2}}-\frac{1}{4R_{s}^{2}\left(s\right)}+\alpha \tilde{r}\left(s\right)\cos\left[\vartheta -\varphi \left(s\right)\right]-\frac{2}{\sqrt{{\tilde{r}}_{eD}^{2}\left(s,s_{D}\right)+{\left(z-z_{D}\right)}^{2}}}\\ \alpha =\frac{eFa_{0}^{*}}{R_{y}^{*}};\;{\tilde{r}}_{eD}^{2}\left(s,s_{D}\right)={\left[\tilde{x}\left(s\right)-\tilde{x}\left(s_{D}\right)\right]}^{2}+{\left[\tilde{y}\left(s\right)-\tilde{y}\left(s_{D}\right)\right]}^{2} \end{array}$$

The four terms in the potential energy (Equation 3) correspond to the following contributions: from the lateral confinement inside the nanolayer, from the curvature variation along the spiral [26,27], from the external electric field and from the electron–donor attraction, respectively. In Figure 2, we show curves of potential energies given by the relations (Equation 1)–(Equation 3) for the Archimedean spiral with parameters $a=b=2a_{0}^{*}$, $\varphi_{F}=10\pi$ and with the donor located at the point with polar coordinate $\varphi_{D}=4\pi$. The solid line corresponds to the zero-electric field case, while the dash line corresponds to the case F=2 kV/cm. One can see in Figure 2 an essential change of the curve of the potential energy, which, in the presence of the external electric field, obtains a form of oscillation with increasing amplitude along the spiral arc. Such modification of the potential can induce, as we show below, a strong polarizability and the formation of a giant dipole moment in the ground state of the shallow donor. To explain how the increasing external electric field applied along the *X*-axis can induce a giant permanent moment in a donor, we display in Figure 3 some plots for the electron (a) and for the donor (b), which show a successive evolution of the potential curves along the spiral at the cross section z=0.

The curve of the free electron’s potential in Figure 3a for the zero electric field case (solid line) is smooth, and it has a shallow minimum at the initial point of the spiral (s=0), where the curvature of the spiral is the largest. In the presence of the external electric field directed toward the initial spiral point along the *X*-axis, this potential is transformed to a set of periodically arranged quantum wells with successively increasing depth at the radial direction (dash and dot lines). The bottoms and tops of quantum wells are located at points of corresponding wings with polar coordinates $\varphi_{k}=\left(2k+1\right)\pi$ and $\varphi_{k}=2k\pi$, respectively. As the electric field grows, the bottoms of all wells descend, while their heights are increased. It is seen that the larger the electric field, the bigger the displacement of the electron should be from the initial point of the spiral.

Similar potential curves for the donor placed inside RUQW at the position with polar angle are presented in Figure 3b. As F=0 (solid line), the main deepest minimum is situated at the donor location, while other minima with less depth are located at points of nearest wings with polar angles $\varphi_{k}=2k\pi$, situated at the same side of spiral. With an increase in the electric field, additional minima appear on both sides of the main minimum at points with polar angles $\varphi_{k}=\left(2k-1\right)\pi$, at which the minima on the right side are deeper than those on the left. The larger the electric field, the deeper the additional quantum wells (cf. dashed and dotted curves). One can expect that such an alteration of the potential curves under increasing electric field could lead to an instability of the donor ground state when the energy of a similar state of one of the additional minima at the right side becomes lower than that of the main minimum, and the electron jumps from the wing with the donor toward the right side of the spiral. Such a transition can be accompanied by an origination of a giant dipole moment and a strong polarizability of the donor. The final 2D wave equation in dimensionless units for the electron inside a two-dimensional surface with the potential energy (Equation 3) has, in natural coordinates, the following form [7]:(4)$$\begin{array}{c} -\frac{\partial^{2}\Phi \left(s,z\right)}{\partial s^{2}}-\frac{\partial^{2}\Phi \left(s,z\right)}{\partial z^{2}}+V\left(s,z\right)\Phi \left(s,z\right)=E\left(s,z\right)\Phi \left(s,z\right);\;0<s<s_{F};\;0<z<L\\ \Phi \left(0,z\right)=\Phi \left(s_{F},z\right)=\Phi \left(s,0\right)=\Phi \left(s,L\right) \end{array}$$

Here we assume that $z_{D}=L/2$, i.e., the donor is located in the middle along the *Z*-axis of the structure, which allows to depreciate the effect of the surface states. In our numerical work, we solve the boundary value problem (Equation 4) by using the double Fourier series expansion method, in which the wave function is represented as follows:(5)$$\Phi \left(s,z\right)=\sum_{n,m=-\infty }^{\infty }C_{n,m}\chi_{m,n}\left(s,z\right);\;\chi_{m,n}\left(s,z\right)=\frac{2}{\sqrt{S_{F}L}}\sin\frac{\pi ms}{S_{F}}\sin\frac{\pi nz}{L};\;n,m=1,2,\ldots ;$$

The functions $\chi_{m,n}\left(s,z\right)$ present an orthogonal and normalized basis, which satisfy the boundary conditions. Substituting (Equation 5) in (Equation 4), one can reduce the eigenvalue problem (Equation 4) to the following secular equation:(6)$$\begin{array}{c} \sum_{n,m=-\infty }^{\infty }\left(H_{m,m\prime ,n,n\prime }-\tilde{E}\delta_{m,m\prime }\delta_{n,n\prime }\right)C_{m\prime ,n\prime }=0;\\ H_{m,m\prime ,n,n\prime }=\pi^{2}\left(\frac{m^{2}}{S_{F}^{2}}+\frac{n^{2}}{L^{2}}\right)\delta_{m,m\prime }\delta_{n,n\prime }+U_{m,m\prime }^{\left(1\right)}\delta_{n,n\prime }+U_{m,m\prime ,n,n\prime }^{\left(2\right)} \end{array}$$
(7)$$U_{m,m\prime }^{\left(1\right)}=\;\frac{2}{S_{F}}\;\int_{0}^{S_{F}}\;\left[\frac{\pi }{d^{2}\left(s\right)}-\frac{1}{4R_{s}^{2}\left(s\right)}+\alpha \tilde{r}\left(s\right)\left[\cos\vartheta \cos\tilde{\varphi }\left(s\right)-\sin\vartheta \sin\tilde{\varphi }\left(s\right)\right]\right]\sin\frac{\pi ms}{S_{F}}\sin\frac{\pi m\prime s}{S_{F}}ds$$
(8)$$U_{m,m\prime ,n,n\prime }^{\left(2\right)}=\;\frac{4}{S_{F}L}\;\int_{0}^{S_{F}}\;\sin\frac{\pi ms}{S_{F}}\sin\frac{\pi m\prime s}{S_{F}}ds\int_{0}^{L}\;\frac{2}{\sqrt{{\tilde{r}}_{eD}^{2}\left(s,s_{D}\right)+{\left(z-L/2\right)}^{2}}}\sin\frac{\pi nz}{L}\sin\frac{\pi n\prime z}{L}dz$$

## 3. Results

In this section, we present the results of the analysis of the effect of the electric field on the energy spectrum of donors confined in a GaAs RUQW of the height $L=50a_{0}^{*}\approx 500\;$nm, and the homogeneous thickness $d=0.5a_{0}^{*}\approx 5\;$nm, whose cross section along the plane perpendicular to the axis Z is the finite Archimedean spiral, given in polar coordinates by the relation r=a+bφ, with $a=b=2a_{0}^{*}\approx 20\;$nm, $0<\varphi <\varphi_{F}$ and $\varphi_{F}=10\pi$, $S_{F}=s\left(\varphi_{F}\right)\approx 220a_{0}^{*}\approx 2200\;$nm. The shifting and splitting of spectral lines of the donor in the presence of an external electric field applied along the XOY plane is expected to be anisotropic due to the lack of axial symmetry in the RUQW structure. Therefore, we show below the results of calculations for different angles ϑ between the electric field vector and X axis, (0≤ϑ≤π) assuming that the donor position inside the spiral is given by cylindrical coordinates ($\varphi_{D}=6\pi ,z_{D}=L/2$).

In Figure 4, we display the lower energy dependencies on the electric field applied along the *X* axis (ϑ=π), which reveal significant differences in the corresponding curves for the electron and the donor. In contrast to the free electron’s energy spectrum in Figure 4a, which is quasi-continuous, the corresponding spectrum of the donor in Figure 4c is split in bands with decreasing gaps between them as the band number grows. Such a structure of the energy spectrum is typical for the electron states in a Coulomb potential field. Additionally, one can observe in Figure 4a,b multiple crossovers of the curves corresponding to the lower energies. It is interesting to note that at the point of crossover of the curves in Figure 4a,b, the slope of the ground state energy dependency on the electric field undergoes a sharp jump off when the increasing electric field attaches to the value of about 0.1kV/cm. Taking into account that this slope defines the dipole moment of the ground state, one can believe that the increase in the slope at this point corresponds to the rearrangement of the ground state configuration due to an abrupt transfer of the electron at a neighboring wing of the spiral.

It is seen that discontinuities between energies $E_{k}\;\left(k=1,2,\ldots \right)$ inside the subbands, both for the electron in Figure 4a and the donor in Figure 4c, are very small, and therefore, it is more appropriate to have the description of the energy spectrum in terms of the density of the electronic states (DOS), which we define as follows:(9)$$\rho \left(E\right)=\frac{1}{\pi }\sum_{k}\frac{\Gamma }{{\left(E-E_{k}\right)}^{2}+\Gamma^{2}}$$

Here, Γ is the phenomenological natural linewidth for the model of the Lorentzian line shape function [27].

In Figure 5, we show curves of the DOS for different values of the electric field for the electron and donor. In both cases, one can observe a consecutive broadening of the principal peak of the DOS with increasing the electric field. However, for the donor, this broadening is almost symmetrical in contrast to the electron, for which the broadening of the peak is accompanied by its shifting to the red end of the spectrum. In addition, in the case of the electron, the increase of the electric field induces the fine structure in the curve of the DOS.

Presented above in Figure 4b, donor energy dependencies on the external electric field show the existence of a critical point, in which their slopes are changed abruptly. Such a feature in the behavior of the curves should be related to a jump of the electron between two adjacent wings of the spiral, induced by the external electric field. The change in the configuration of the donor ground state should be also accompanied by a sharp variation of the dipole moment *p* and the polarizability α of the donor, related to the first and the second derivatives of the ground-state energy as a function of the external electric field [28]:(10)$$p\left(F\right)=-\frac{dE_{0}\left(F\right)}{dF};\;\alpha \left(F\right)=\frac{dp(F)}{dF}=\frac{d^{2}E_{0}\left(F\right)}{dF^{2}}$$

In Figure 6a, we display the plot of the averaged separation between the electron and donor $\langle r_{eD}\rangle =p/e$ as a function of the external electric field *F*. One can see that with an increase in the electric field above the value of 0.04kV/cm, the averaged electron–donor separation begins to increase sharply until it reaches the largest value for the electric field of 0.08kV/cm. We attribute such dependence of the dipole moment to a rather sharp change in the configuration of the ground state of the donor.

One can associate the small value of the dipole moment for fields below 0.04kV/cm with the configuration, in which the electron is located mainly on the same wing of the helix as the donor. For fields between 0.04kV/cm and 0.08kV/cm, the electron is shifted toward the adjacent wing and is located mainly on opposite sides of the spiral with a significantly increased separation with the donor. In addition, the plot presented in Figure 6b for the dependence of the polarizability of the donor ground state on the external electric field confirms such an interpretation. One can observe a giant polarizability of the donor ground state in the presence of the external electric fields between 0.04kV/cm and 0.08kV/cm.

The Stark effect in donors confined in the RUQW is not only nonlinear but also is anisotropic, i.e., the shifting and splitting of energy levels depend not only on the value of the external field but also on its direction. It is related to the spiral’s geometry, in which the separation between adjacent wings at each point depends not only on the point position but also on the direction. Finally, in Figure 7, we present the energies of the donor as a function of the angle θ between the electric field direction and *X* axis. It is seen that for a weak electric field, as the donor dipole moment is small, the lower energies in Figure 7a almost do not depend on the electric field orientation because these states are strongly bound to the donor. The behavior of the lower levels in Figure 7b is essentially changed, as the electric field is equal to 0.1kV/cm and the electron–donor separation is large according to the results presented in Figure 6a. The energies of the ground and of first excited states become sensitive to the orientation of the electric field in contrast to the corresponding dependencies in Figure 7a.

## 4. Conclusions

Low-dimensional systems based on semiconductor materials have attracted great interest in recent decades. The quantum properties of these systems make it possible to control and change physical behavior at the nanoscale, thus opening up many new possible applications. However, a greater diversity of spectral properties could be expected from donors confined in rolled-up nanofilms with variable curvature. In particular, the donor confined in such a structure can, strictly speaking, no longer be considered neither a two-dimensional hydrogen-like atom nor a three-dimensional one due to the possible long-range interaction between the electron and donor situated on different wings of the spiral. In this work, we propose a simple computational procedure, which allows us to analyze the Stark shifting and splitting of spectral lines of donors added into spirally rolled-up quantum wells due to the presence of an external electric field, applied at a direction perpendicular to the spiral axis.

Considering, as an example, an Archimedean shape of the spiral, we investigated theoretically the effect of its curvature, the position of a donor, the intensity, and the orientation of the external electric field on the donor energy levels. We present novel curves for the electron and the donor energies dependencies on the value *y* orientation of the electric field. We analyzed the corresponding transformations of the densities of state curves.

We found two specific features of the Stark effect in donors added into a spirally rolled-up quantum well. First, the Stark shifting of donor energy levels is strongly nonlinear, and it is accompanied by an abrupt change in the dipole moment and polarizability of the donor associated with the electron’s jump between neighboring wing spirals. Second, the Stark effect in donors placed into spirally rolled-up quantum wells is essentially anisotropic, the shifting and the splitting of the donor energy levels depending on the orientation of the electric field.

Our results show that the spectral properties of rolled-up quantum wells can be essentially modified by varying geometrical parameters. We believe that our computational method can be applicable to many complex quasi-two-dimensional structures with curvature, for which more rigorous approaches require extensive numerical calculations.

## Figures and Tables

**Figure 1 micromachines-14-01290-f001:**
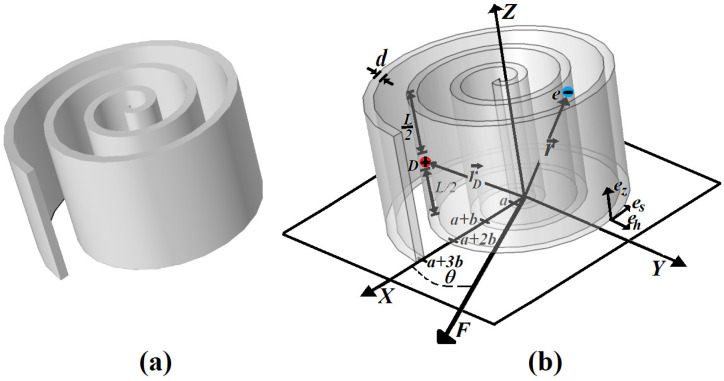
Characteristics of the model: (**a**) Image of a rolled-up quantum well considered in this work, (**b**) 3D image of donor confined in a rolled-up QW, where the electron and donor positions, and the orientation of applied electric field are indicated.

**Figure 2 micromachines-14-01290-f002:**
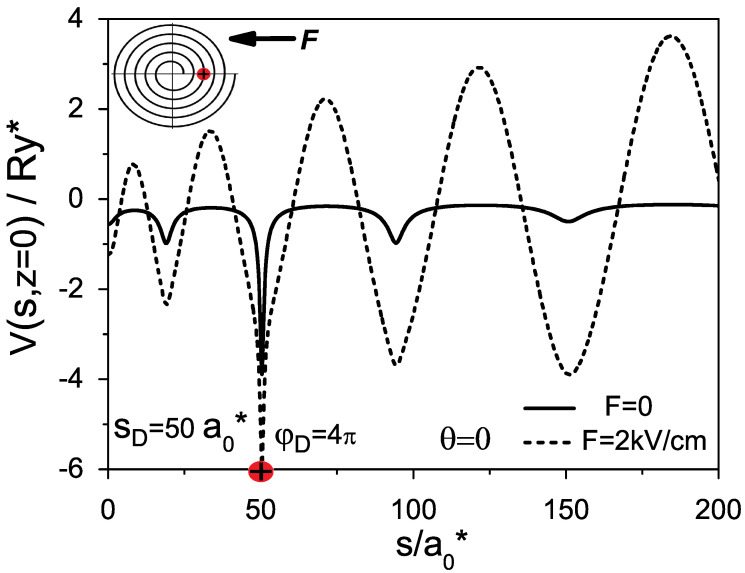
Alteration of the donor’s potential energy along the spiral, under external electric field.

**Figure 3 micromachines-14-01290-f003:**
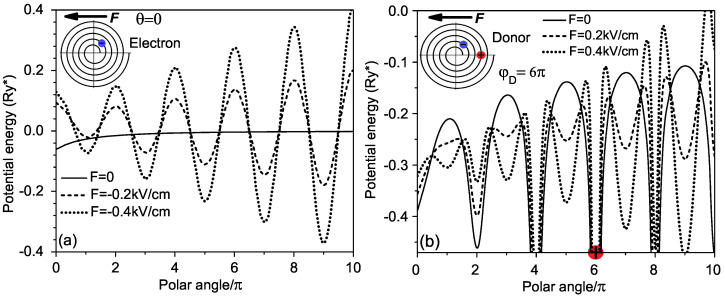
Potential energies along the spiral at the cross section z=0, for (**a**) the electron and (**b**) the donor located at the position with polar coordinate $\varphi_{D}=6\pi$, for different values of the external electric field applied along *X*-axis.

**Figure 4 micromachines-14-01290-f004:**
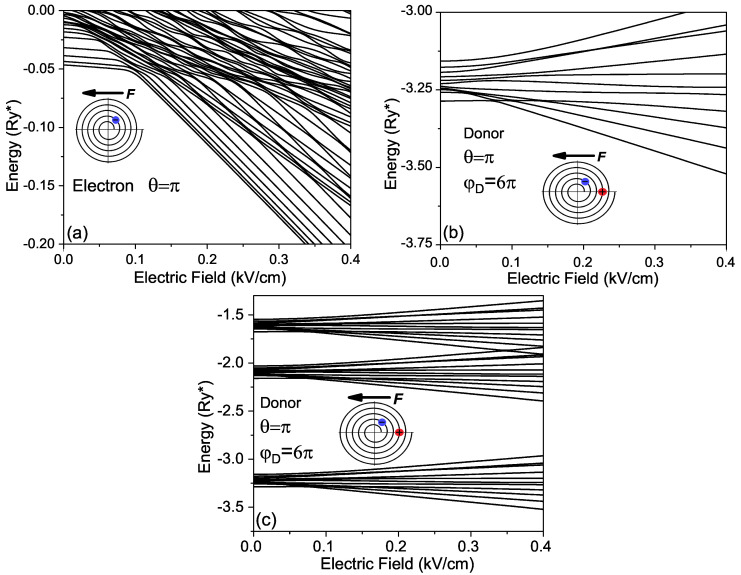
Lowest energies of (**a**) electron, (**b**) donor and (**c**) some excited states of donor in rolled-up QW as functions of the external electric field.

**Figure 5 micromachines-14-01290-f005:**
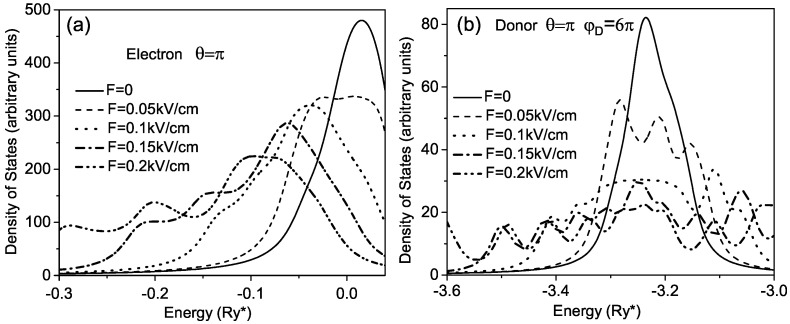
Density of states for (**a**) electron and (**b**) donor in rolled-up QW for different values of the electric field.

**Figure 6 micromachines-14-01290-f006:**
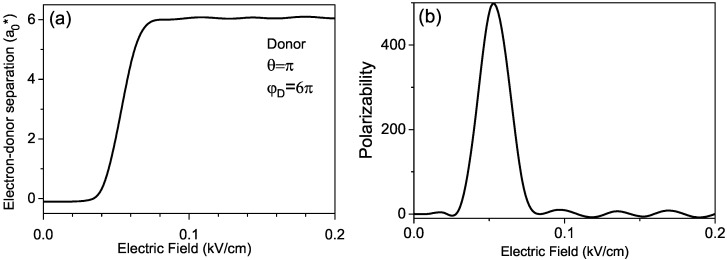
Dependencies of (**a**) the averaged electron–donor separation and (**b**) the donor polarizability on the external electric field.

**Figure 7 micromachines-14-01290-f007:**
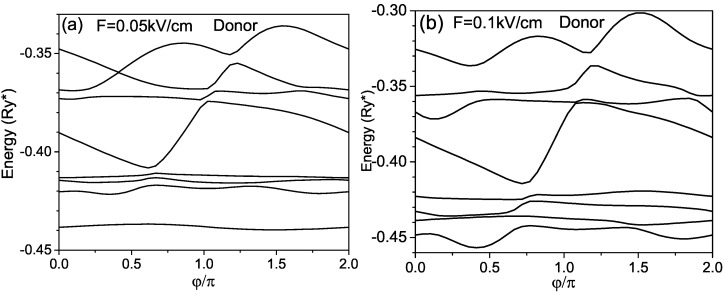
Energies of donor in rolled-up QW as functions of the angle applied electric field.

## Data Availability

No new data were created or analyzed in this study. Data sharing is not applicable to this article.

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
