# Peer review of "Stark Effect for Donors in Rolled-Up Quantum Well"

_micromachines, 2023, doi:10.3390/mi14071290_

Round 1

Reviewer 1 Report

The rolled-up structure is a versatile and powerful tool in nanotechnology, with a wide range of potential applications in electronics and other areas. It's important to understand the electron behaivor in rolled-up structure. This manuscript calculate the energy spectra of shallow donors in a rolled-up quantum well with external electrical field. I have some questions regarding the calculation.

1. The assumption made in this manuscript's model is that the structure along the Z-axis is significantly longer than its diameter. At what diameter-to-length ratio does this assumption become invalid?

2. In a rolled-up structure, electrons can move in the XOY plane either by following the curvature of the structure or by directly tunneling between adjacent spiral wings. If an electron or hole moves from one spiral wing to another, is it possible to determine the probability of direct tunneling versus movement along the curvature of the spiral?

Reviewer 2 Report

The text is very convoluted. There are many long sentences that make the text very hard to grasp. The authors must rework the draft so it becomes more accessible, and clearer.

Round 2

Reviewer 2 Report

The authors are appreciated for improving the readability of the manuscript.

The authors did try very hard to ratify the written english. The text reads much better than the previous version. Good work!